

# Constructing a prognostic tool for predicting the risk of non-adherence to antiplatelet therapy in discharged patients with coronary heart disease: a retrospective cohort study

Jiaoyu Cao[*], Lixiang Zhang[*] and Xiaojuan Zhou

Department of Cardiology, The First Affiliated Hospital of the University of Science and Technology of China, Hefei, China

[*] These authors contributed equally to this work.

Corresponding author
Jiaoyu Cao, caojiaoyu@126.com

**OPEN ACCESS**

## ABSTRACT

**Objective**. To investigate the incidence and influencing factors affecting the non-adherence behavior of patients with coronary heart disease (CHD) to antiplatelet therapy after discharge and to construct a personalized predictive tool.

**Methods**. In this retrospective cohort study, 289 patients with CHD who were admitted to the Department of Cardiology of The First Affiliated Hospital of the University of Science and Technology of China between June 2021 and September 2021 were enrolled. The clinical data of all patients were retrospectively collected from the hospital information system, and patients were followed up for 1 year after discharge to evaluate their adherence level to antiplatelet therapy, analyze their present situation and influencing factors for post-discharge adherence to antiplatelet therapy, and construct a nomogram model to predict the risk of non-adherence.

**Results**. Based on the adherence level to antiplatelet therapy within 1 year after discharge, the patients were divided into the adherence ($n = 216$) and non-adherence ($n = 73$) groups. Univariate analysis revealed statistically significant differences between the two groups in terms of variable distribution, including age, education level, medical payment method, number of combined risk factors, percutaneous coronary intervention, duration of antiplatelet medication, types of drugs taken at discharge, and CHD type ($P < 0.05$). Furthermore, multivariate logistic regression analysis revealed that, except for the medical payment method, all the seven abovementioned variables were independent risk factors for non-adherence to antiplatelet therapy ($P < 0.05$). The areas under the receiver operating characteristic curve before and after the internal validation of the predictive tool based on the seven independent risk factors and the nomogram were 0.899 (95% confidence interval [CI]: 0.858–0.941) and 0.89 (95% CI: 0.847–0.933), respectively; this indicates that the tool has good discrimination ability. The calibration curve and Hosmer–Lemeshow goodness of fit test revealed that the tool exhibited good calibration and prediction consistency ($\chi^2 = 5.17$, $P = 0.739$).

**Conclusion**. In this retrospective cohort study, we investigated the incidence and influencing factors affecting the non-adherence behavior of patients with CHD after discharge to antiplatelet therapy. For this, we constructed a personalized predictive tool based on seven independent risk factors affecting non-adherence behavior.
The predictive tool exhibited good discrimination ability, calibration, and clinical applicability. Overall, our constructed tool is useful for predicting the risk of non-adherence behavior to antiplatelet therapy in discharged patients with CHD and can be used in personalized intervention strategies to improve patient outcomes.

## INTRODUCTION

A recent study have reported that the incidence of coronary heart disease (CHD) in China is increasing every year, with a trend of occurring in young people, seriously endangering human health (*Xiaoyun, 2019*). The treatment strategies for CHD include drug usage, lifestyle changes, interventions, and surgical treatment; among them, percutaneous coronary intervention (PCI) is the primary treatment strategy for CHD at present. With the recent popularization and application of PCI in clinical settings, the prognosis of patients with CHD has considerably improved, and the coronary restenosis rate notably reduced (*Wang et al., 2019*). However, to prevent the occurrence of adverse events after PCI, including stent thrombosis and restenosis, patients need long-term medications (*Hua, 2019*; *Sousa-Uva et al., 2019*; *Savic et al., 2019*).

In China, while the antiplatelet medication adherence of patients with CHD after discharge is generally low, this adherence further decreases with the gradual extension of the post-discharge period (*Liu et al., 2020*). Therefore, it is vital to investigate the present situation of antiplatelet medication non-adherence of patients with CHD after discharge, integrate the non-adherence-associated risk factors, screen patients with high risk of medication non-adherence, and provide targeted intervention programs for such patients to improve their medication adherence and PCI efficacy. However, presently, most studies on the antiplatelet medication adherence of patients with CHD after discharge have focused on the analyses of the current situation and influencing factors (*Bai et al., 2021*; *Yan et al., 2021*; *Pogosova et al., 2021*; *Mozaffarian et al., 2022*; *Prami et al., 2016*), and only a few studies have integrated risk factors and personalized prediction of the risk of the medication non-adherence behavior of patients with CHD who underwent antiplatelet therapy.

In this retrospective cohort study, we analyzed the clinical data of the patients. Furthermore, based on the risk factors associated with non-adherence to antiplatelet therapy in patients with CHD within 1 year after discharge, we constructed a predictive tool based on the nomogram model for the personalized prediction of the risk of non-adherence to antiplatelet therapy in patients with CHD after discharge. This prediction tool was used to screen patients at high risk of medication non-adherence after discharge and strategize targeted intervention measures to improve the medication adherence behavior of patients.
## PATIENTS AND METHODS

### Subjects

In this retrospective cohort study, we enrolled 289 patients with CHD who were admitted to the Department of Cardiology of The First Affiliated Hospital of the University of Science and Technology of China, Anhui, China, from June 2021 to September 2021. The inclusion criteria were as follows: patients meeting the diagnostic criteria of the guidelines for the diagnosis and treatment of CHD issued by the American College of Cardiology/American Heart Association (*Adams, Bojara & Schunk, 2017*) and who were diagnosed with CHD for the first time *via* coronary angiography; patients with clear consciousness and normal cognitive and communication abilities; and patients who were regularly taking antiplatelet drugs during hospitalization. The exclusion criteria were as follows: patients with abnormal communication and cognitive abilities; those who were previously diagnosed and readmitted; and those lost to follow-up after discharge. The study protocol was approved by the Medical Ethics Committee of The First Affiliated Hospital of the University of Science and Technology of China, Anhui, China (ID: 2023-RE-028). It adhered to the principles of the Declaration of Helsinki. Owing to the retrospective nature of the study and anonymized data analysis, informed consent from patients was exempted.

### Method

The demographic characteristics, disease-related data, and medications at discharge were retrospectively obtained from the hospital information system and electronic medical record system. The patients were followed up at 1, 3, and 6 months and 1 year after discharge. Adherence to antiplatelet therapy was evaluated based on telephonic follow-up and outpatient return visits. Similar to previous studies (*Sanfélix-Gimeno et al., 2013*; *Zhu et al., 2011*), the proportion of days covered (PDC) was used to evaluate the drug adherence of patients 1 year after discharge. PDC refers to the number of days covered by medication among the total number of days in the post-discharge follow-up period. Patient adherence to antiplatelet therapy after discharge was defined as follows: (1) patients who underwent PCI and received dual antiplatelet therapy (aspirin and clopidogrel) and had a PDC of ≥80% for each drug; and (2) PDC of ≥80% for monotherapy with aspirin or clopidogrel in patients who did not undergo PCI. Adherence was defined as meeting the abovementioned PDC criteria regarding the medication, and non-adherence was defined as failing to meet the above criteria.

### Statistical analyses

R software (R 3.6.1; *R Core Team, 2019*) was used for data analyses. Measurement data conforming to normal distribution were expressed as mean ± standard deviation. The independent sample $t$-test and Pearson's chi-squared test were performed for comparison between the two groups. Count data were expressed as the number of cases and percentage. Through multivariate logistic regression analysis, the independent influencing factors were screened. The "rms" package of R software was used to plot the nomogram. The receiver operating characteristic(ROC) curve of the nomogram was plotted using the "pROC" package, and the area under the ROC curve (AUC) was used to evaluate the discrimination

of the nomogram model. The calibration and prediction consistency of the model were evaluated using the Hosmer–Lemeshow test for goodness of fit and calibration curve (CR), and the clinical applicability of the nomogram model was evaluated *via* decision curve analysis (DCA). A *p*-value of <0.05 was considered statistically significant.

# RESULTS

## Description of clinical data of subjects

There were 91 (31.49%) females and 198 (68.51%) males. The average age of patients was 59.24 ± 9.97 (31–73) years. Information on education levels was as follows: 166 (57.44%) patients, junior middle school or below; 82 (28.37%) patients, senior middle school or technical secondary school; and 41 (14.19%) patients, college or above. Information on medical payment methods was as follows: 37 (12.8%) patients, self-paid; 140 (48.44%) patients, resident's medical insurance; and 112 (38.75%) patients, employee's medical insurance. The numbers of patients with corresponding combined risk factors were as follows: 49 (16.96%) patients, zero factors; 96(33.22%) patients, one factor; 58 (20.07%) patients, two factors; 52(17.99%) patients, three factors; and 34 (11.76%) patients, four factors. Among the 289 patients, 243 patients (84.08%) received PCI. The patients had the following types of CHD: 97 (33.56%) patients, stable angina pectoris (SAP); 137 (47.4%) patients, unstable angina pectoris; and 55 (19.03%) patients, acute myocardial infarction (AMI). The duration of antiplatelet drug administration was as follows: <2 years, 78 (26.99%) patients; 2–5 years, 116 (40.14%) patients; and >5 years, 95 (32.87%) patients. Detailed demographic data are presented in Table 1.

## Univariate analysis of non-adherence to antiplatelet therapy in patients with CHD

Among the 289 patients with CHD, 73 (25.26%) exhibited non-adherence to antiplatelet therapy within 1 year after discharge. Statistically significant differences in the medication adherence rate of patients were observed owing to the following eight variables: age, medical payment method, educational level, number of combined risk factors, PCI, duration of antiplatelet drug administration, type of CHD, and types of drugs taken after discharge ($P < 0.05$) (Table 1).

## Multivariate logistic regression analysis of non-adherence to antiplatelet therapy in patients with CHD

Univariate and multivariate logistic regression analyses were performed with the abovementioned eight variables, which revealed statistically significant differences in the variables preliminarily screened in Table 1 (independent variables) and the drug adherence of patients within 1 year after discharge, which was the dependent variable. Based on the results of univariate and multivariate logistic regression analyses, age, education level, number of combined risk factors, PCI, course of antiplatelet medication, type of CHD, and types of medication taken at discharge were determined as the independent risk factors for the occurrence of antiplatelet medication non-adherence ($P << 0.05$) (Table 2).

**Table 1** Univariate analysis results of antiplatelet medication adherence.

| Variables | Total ($n = 289$) | Adherence group ($n = 216$) | Non-adherence group ($n = 73$) | $t/\chi^2$ | $P$ |
|---|---|---|---|---|---|
| Age | 59.24 ± 9.97 | 58.06 ± 10.42 | 62.71 ± 7.55 | −4.108[a] | <0.001 |
| BMI | 21.44 ± 3.16 | 21.24 ± 3.25 | 22.03 ± 2.80 | −1.844[a] | 0.066 |
| Sex | | | | 0.083[b] | 0.774 |
| Female | 91 (31.49%) | 69 (31.94%) | 22 (30.14%) | | |
| Male | 198 (68.51%) | 147 (68.06%) | 51 (69.86%) | | |
| Educational level | | | | 17.046[b] | <0.001 |
| Junior high school and below | 166 (57.44%) | 109 (50.46%) | 57 (78.08%) | | |
| Senior high school and technical secondary school | 82 (28.37%) | 71 (32.87%) | 11 (15.07%) | | |
| College degree or above | 41 (14.19%) | 36 (16.67%) | 5 (6.85%) | | |
| Medical payment method | | | | 17.249[b] | <0.001 |
| At one's own expense | 37 (12.80%) | 29 (13.43%) | 8 (10.96%) | | |
| Resident medical insurance | 140 (48.44%) | 118 (54.63%) | 22 (30.14%) | | |
| Employee medical insurance | 112 (38.75%) | 69 (31.94%) | 43 (58.90%) | | |
| Number of combined risk factors | | | | 99.556[b] | <0.001 |
| 0 | 49 (16.96%) | 45 (20.83%) | 4 (5.48%) | | |
| 1 | 96 (33.22%) | 91 (42.13%) | 5 (6.85%) | | |
| 2 | 58 (20.07%) | 49 (22.69%) | 9 (12.33%) | | |
| 3 | 52 (17.99%) | 20 (9.26%) | 32 (43.84%) | | |
| 4 | 34 (11.76%) | 11 (5.09%) | 23 (31.51%) | | |
| PCI or not | | | | 14.757[b] | <0.001 |
| No | 46 (15.92%) | 24 (11.11%) | 22 (30.14%) | | |
| Yes | 243 (84.08%) | 192 (88.89%) | 51 (69.86%) | | |
| Type of coronary heart disease | | | | 8.738[b] | 0.013 |
| SAP | 97 (33.56%) | 79 (36.57%) | 18 (24.66%) | | |
| UAP | 137 (47.40%) | 104 (48.15%) | 33 (45.21%) | | |
| AMI | 55 (19.03%) | 33 (15.28%) | 22 (30.14%) | | |
| Occupational status | | | | 0.775[b] | 0.379 |
| Unemployed or retired | 217 (75.09%) | 165 (76.39%) | 52 (71.23%) | | |
| On the job | 72 (24.91%) | 51 (23.61%) | 21 (28.77%) | | |
| Type of drugs taken at discharge | | | | 10.256[b] | 0.001 |
| 1–2 drugs | 113 (39.10%) | 96 (44.44%) | 17 (23.29%) | | |
| ≥3 | 176 (60.90%) | 120 (55.56%) | 56 (76.71%) | | |
| Duration of antiplatelet medication | | | | 26.703[b] | <0.001 |
| <2 years | 78 (26.99%) | 70 (32.41%) | 8 (10.96%) | | |
| 2–5 year | 116 (40.14%) | 92 (42.59%) | 24 (32.88%) | | |
| >5 years | 95 (32.87%) | 54 (25.00%) | 41 (56.16%) | | |

**Notes.**
[a]Independent sample $t$-test
[b]Pearson's chi-squared test
SAP, stable angina pectoris; UAP, unstable angina pectoris; AMI, acute myocardial infarction.

**Table 2 Results of logistic regression analysis of the risk factors for non-adherence among patients with oral medication 1 year after discharge.**

| Variables | Univariate logistic regression analysis | | Multivariate logistic regression analysis | |
|---|---|---|---|---|
| | OR (95% CI) | P | OR (95% CI) | P |
| Age | 1.06 (1.02–1.10) | <0.001 | 1.06 (1.01–1.10) | 0.018 |
| Educational level | 0.42 (0.27–0.67) | <0.001 | 0.55 (0.31–0.98) | 0.041 |
| Medical payment method | | | | |
| At one's own expense | Ref | | Ref | |
| Resident medical insurance | 0.68 (0.27–1.67) | 0.396 | 0.30 (0.09–1.02) | 0.054 |
| Employee medical insurance | 2.26 (0.95–5.39) | 0.067 | 1.35 (0.42–4.34) | 0.616 |
| Number of combined risk factors | 3.18 (2.37–4.26) | <0.001 | 3.48 (2.40–5.03) | <0.001 |
| PCI or not | | | | |
| No | Ref | | Ref | |
| Yes | 0.29 (0.15–0.56) | <0.001 | 0.36 (0.15–0.87) | 0.023 |
| Type of coronary heart disease | | | | |
| SAP | Ref | | Ref | |
| UAP | 1.39 (0.73–2.65) | 0.314 | 2.14 (0.88–5.21) | 0.092 |
| AMI | 2.93 (1.39–6.16) | 0.005 | 3.37 (1.17–9.70) | 0.024 |
| Type of drugs taken at discharge | | | | |
| 1–2 drugs | Ref | | Ref | |
| ≥3 | 2.64 (1.44–4.83) | 0.002 | 2.28 (1.01–5.11) | 0.046 |
| Duration of antiplatelet medication | 2.66 (1.79–3.95) | <0.001 | 2.05 (1.20–3.51) | 0.008 |

**Notes.**

SAP, stable angina pectoris; UAP, unstable angina pectoris; AMI, acute myocardial infarction.

## Nomogram model construction

Based on the results of multivariate logistic regression analysis, a nomogram model was constructed for predicting the risk of non-adherence to antiplatelet therapy in patients with CHD 1 year after discharge using the "rms" package (Fig. 1). Nomogram interpretation involved several steps. First, a vertical line was drawn upward on the horizontal axis of each predictor variable in the nomogram, allowing the determination of the specific scores corresponding to the horizontal axis of "Point". Next, the scores from each predictor variable were added together to obtain a total score, which was then drawn on the horizontal axis of "Total Point" with a downward vertical line. Finally, the corresponding risk values were obtained on the horizontal axis of "Risk of medication non-adherence" as the predicted probability of the nomogram model.

## Analysis of the clinical applicability of the nomogram model

Figure 2 demonstrates the plotted DCA curve of the nomogram model. According to the DCA curve, when the threshold probability of non-adherence was 0.04–0.86, compared with the "full intervention" and "no intervention" programs, the nomogram model provided a better net clinical benefit to patients, suggesting its good clinical applicability.

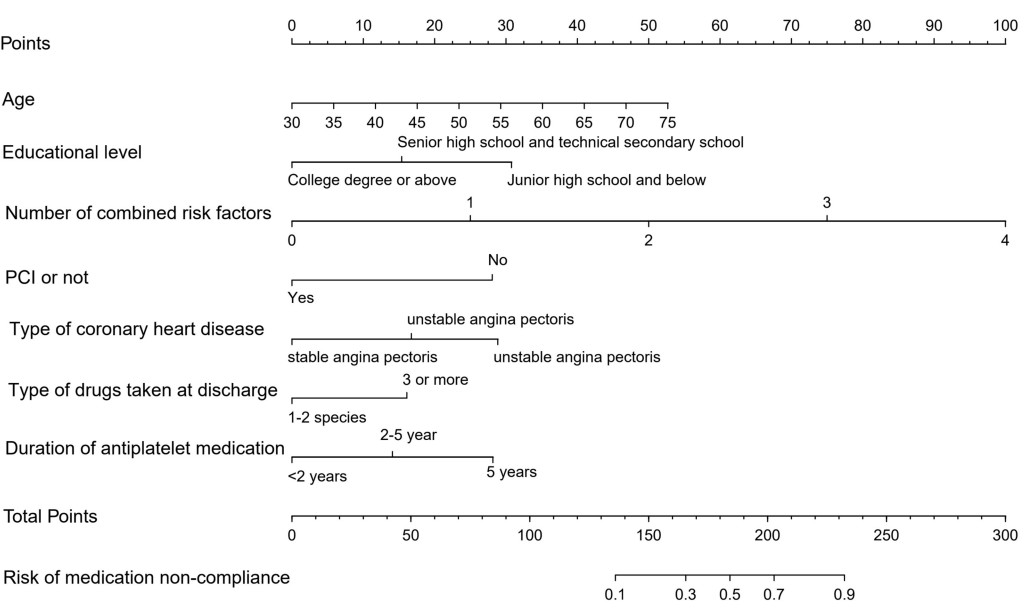

**Figure 1 Nomogram predicting the risk of antiplatelet non-adherence in patients with coronary heart disease.**

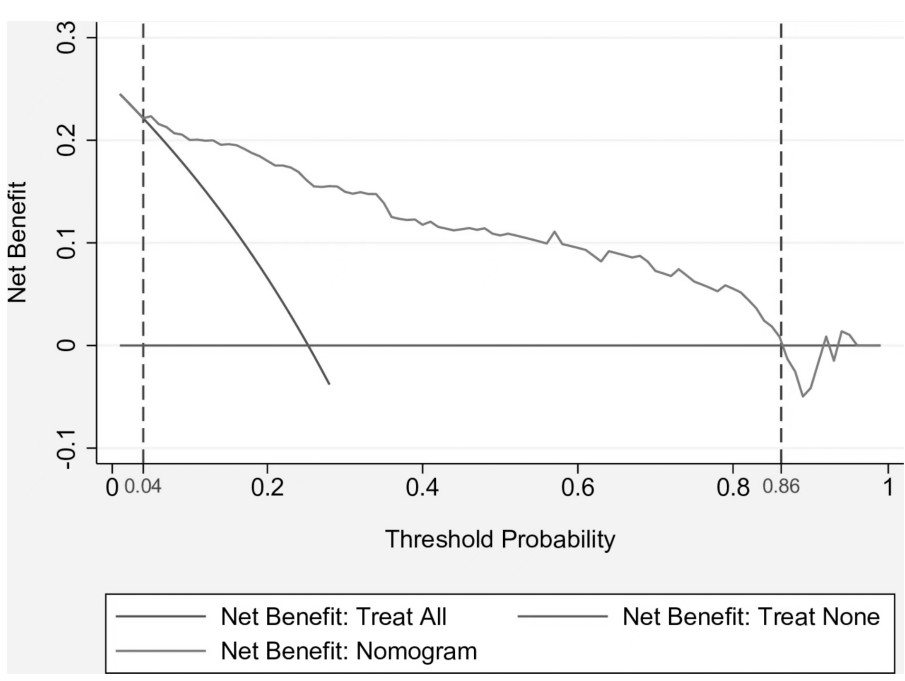

**Figure 2 Clinical decision curve for nomogram model.**

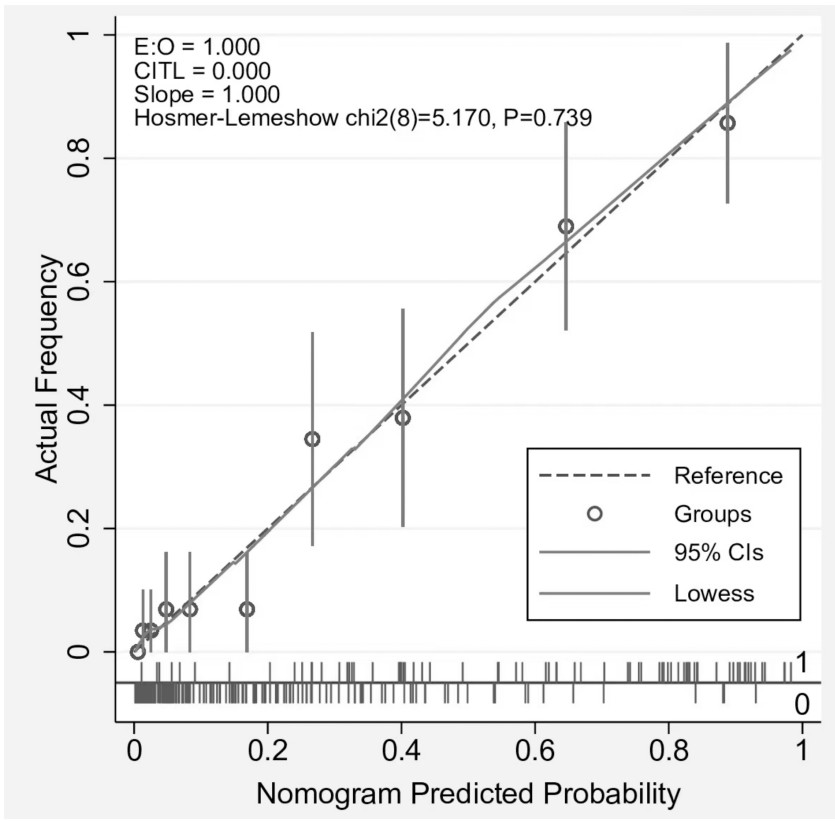

**Figure 3** **Calibration curve for nomogram model.**

### Evaluation of the discrimination and calibration of the nomogram model

The predicted curve of the nomogram model was similar to the ideal curve, as shown in the CR (Fig. 3). The results of the Hosmer–Lemeshow test for goodness of fit suggested that the prediction deviation between the predicted probability of the nomogram model and actual frequency of medication adherence was statistically insignificant ($\chi^2 = 5.17$, $P = 0.739$), indicating the good calibration degree and prediction consistency of the nomogram model. To prevent overfitting of the nomogram model, the bootstrap sampling method (resampling 1000 times) was used for the internal validation of the model to eliminate the effect of overfitting on its predictive stability. The ROC curves of the nomogram model before and after internal validation were plotted using the "pROC" package (Fig. 4). Based on the ROC curve, the AUCs before and after the internal validation of the nomogram model were 0.899 (95% confidence interval [CI]: 0.858–0.941) and 0.89 (95% CI [0.847–0.933]), respectively, indicating the good discrimination ability of the nomogram model.

## DISCUSSION

Drug treatment forms the basis for the prevention and treatment of CHD, with medication adherence being a vital factor affecting drug treatment of CHD (*Yuheng & Lijun, 2019*). Antiplatelet drugs are widely used to prevent and treat CHD; however, many studies have
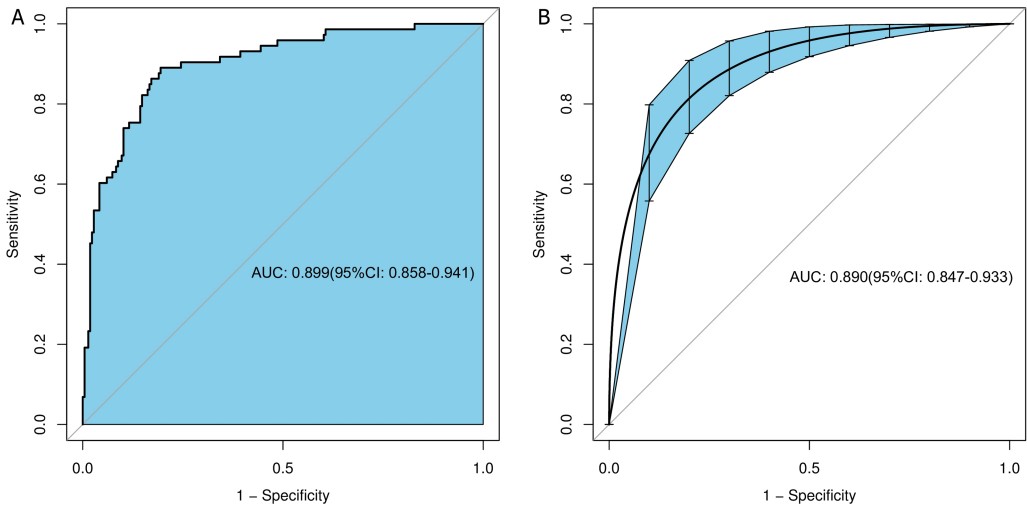

**Figure 4** **ROC curves before and after internal validation of the nomogram model.** (A) ROC curve before internal validation, (B) ROC curve after internal validation.

reported that adherence to antiplatelet drugs among patients with CHD after discharge is poor; *Bai et al. (2021)* and *Yan et al. (2021)* have reported that the antiplatelet drug non-adherence rate among patients with CHD within 1 year after discharge is relatively close, *i.e.,* 17.13% and 17%, respectively. In the present study, the medication non-adherence rate was observed to be 25.26%, which is slightly higher than the abovementioned results but lower than the non-adherence rate of 37% observed in *Yamei (2011)* on dual antiplatelet therapy within 1 year after PCI in patients with CHD. Therefore, targeted prevention and control measures are urgently warranted to decrease antiplatelet drug non-adherence rate after discharge. A study has reported (*Zhang et al., 2015*) that during the development of medication management, the effect of pre-prevention is better than that of in-process and post-intervention. Therefore, it is particularly vital to adopt effective tools for screening patients with CHD who are at high risk of non-adherence behavior after discharge and to implement targeted medication education or other intervention measures in advance.

In the present study, we observed that age, education level, number of combined risk factors, PCI treatment, duration of antiplatelet drugs, CHD type, and types of drugs taken at discharge were independent influencing factors for antiplatelet drug non-adherence in patients with CHD within 1 year after discharge ($P < 0.05$). The limb activity ability and memory of patients gradually decline as age increases, and the decline of activity and memory abilities negatively affects the medication adherence of patients after discharge, thereby decreasing the medication adherence rate (*Wu, 2019*). Patients with high education levels have a relatively high drug literacy level, strong ability to obtain drug information and manage self-medication, and good medication decision-making ability to ensure good medication adherence (*Lai et al., 2020*). Patients with multiple risk factors and drug types are considered to be seriously ill and are administered multiple drugs, resulting in increased memory and drug management burden. On the other hand, patients with multiple diseases

and drug types increase treatment burden and economic load on patients, decrease patient willingness to take drugs (*Fried et al., 2011*; *Jerzak, Pallan & Gerstein, 2014*), indirectly limit the drug adherence of patients, and lead to patients exhibiting drug non-adherence behavior. There are several reasons why patients with CHD who receive PCI tend to have a higher medication adherence rate than those who do not receive PCI. Furthermore, patients with AMI exhibit a higher medication adherence rate than those with SAP. One contributing factor to the higher adherence rate is that these patients, particularly those who receive PCI treatment, incur high treatment costs, possibly leading to increased attention and motivation toward medication belief and adherence behavior. Based on the importance of the prevention of postoperative complications in patients with AMI and/or receiving PCI treatment, the medical staff should have a strong educational background and should be able guide these patients during health education and medication guidance, resulting in these patients having high medication adherence. Simultaneously, in the present study, we observed that a longer medication duration was associated with an increased risk of drug non-adherence behavior among patients, consistent with the findings of *Hu et al. (2021)*, and may be related to the long-term medication of patients, treatment discontinuation after improvement of subjective symptoms, and drug withdrawal without authorization. Healthcare providers should inform patients about the comprehensive drug treatment plan and stress the importance of following the prescribed treatment schedule to ensure optimal outcomes. By communicating this information, patients can better understand the drug treatment regimen and be more inclined to adhere to it throughout their treatment course.

The nomogram model is based on the logistic or Cox regression model, and nomograms are visual representations and intuitive. A nomogram can integrate multiple risk factors, is flexible and practical, and can be used in clinical settings (*Geng, 2018*). Studies have confirmed that the nomogram model can be used to predict long-term adherence to clozapine therapy using neutrophil fluorescence (*Man et al., 2018*), post-discharge safety outcomes in hospitalized patients taking oral warfarin (*Chamoun et al., 2017*), the survival rate of patients after surgery for gastric cancer (*Reim et al., 2015*), and the mortality of patients with chronic heart failure (*Barlera et al., 2013*). However, studies on a nomogram prediction model of the risk of antiplatelet drug non-adherence in patients with CHD after discharge are lacking. As a result, we constructed a nomogram model to predict the risk of non-adherence to antiplatelet therapy in patients with CHD after discharge. The AUC of the nomogram was 0.899 (95% CI [0.858–0.941]) and 0.89 (95% CI [0.847–0.933]) before and after validation, respectively; this indicates that the nomogram has good discrimination ability. The CR and Hosmer–Lemeshow test for goodness of fit revealed that the probability of the nomogram model was similar to the actual frequency of non-adherence to antiplatelet therapy in patients with CHD; however, there was no statistical significance in the prediction deviation between the probability and actual frequency ($\chi^2$ = 5.170, $P$ = 0.739). This suggests that the nomogram model has good calibration and prediction consistency. DCA revealed that when the threshold probability of non-adherence is 0.04–0.86, compared with the "full intervention" and "no intervention" schemes, the nomogram model can help patients achieve a better clinical net benefit; this suggests that the nomogram model has better clinical applicability. Taken together, the nomogram model

constructed based on the influencing factors for antiplatelet medication non-adherence has good prediction efficiency and clinical applicability and can help screen patients at high risk of medication non-adherence after discharge in advance for the medical staff in clinical settings. This can allow the implementation of targeted intervention measures in advance, decrease the occurrence of medication non-adherence among patients, and thereby avoid the occurrence of drug treatment-related adverse events. In addition, the net benefit curve exhibits significant potential value in the context of machine learning, particularly in the development of medicine-related predictive models. By using the net benefit curve, machine learning approaches can be better informed and improved, resulting in more accurate predictions and ultimately achieving better patient benefits. The curve can provide a visual representation of the benefits and costs associated with using a particular model, thereby allowing researchers to optimize their model for maximum net benefit. Overall, the net benefit curve is a valuable tool in the continuous development of machine learning applications in the field of healthcare.

Nevertheless, the present study has some limitations. First, patients living alone and those discharged to family settings should be separately discussed and analyzed, rather than being discussed together with other study subjects as a whole. Second, the importance of age for medication adherence should be emphasized, rather than overlooked. For example, in terms of anticoagulant therapy in patients with atrial fibrillation, older people with atrial fibrillation are at the highest risk of ischemic stroke but are least likely to receive anticoagulant thereby; however, they adhere to this treatment and maintain long-term medication adherence. The reasons for this adherence are multifactorial, including patient-driven factors, physician-driven factors, complexity of the healthcare system, and current lack of knowledge on the biology and natural history of atrial fibrillation (*Hylek, 2020*). The best way to assess the quality of warfarin treatment is using the quality measure time in the treatment range (TTR). Age affects TTR *via* higher medication adherence, and strategies to increase medication adherence may improve the quality of anticoagulant therapy using warfarin (*Marcatto et al., 2016*). Third, the study cohort included patients with CHD at the Department of Cardiology of a hospital in the central region of China; therefore, sample size and representativeness are limited. Finally, unmeasured confounders may have had a potential effect on the study results. Nevertheless, internal validation confirmed that the nomogram does not exhibit an overfitting phenomenon; however, the extrapolation of the nomogram remains unknown. Additional multicenter external validation studies with large samples and more variables will be conducted in the future to determine the prediction stability and extrapolation of the nomogram model.

## CONCLUSION

The incidence of non-adherence to antiplatelet therapy is high in patients with CHD after discharge. Age, education level, number of combined risk factors, duration of antiplatelet drug use, PCI, CHD type, and type of drugs taken at discharge are all independent influencing factors for medication non-adherence in patients with CHD after discharge. The ROC curve, CR, Hosmer–Lemeshow test, and DCA revealed that the constructed

nomogram model exhibits good predictive accuracy and clinical applicability. The proposed model can not only be used to identify patients at high risk of medication non-adherence after discharge but is also useful for targeted medication health education for these patients. Better secondary prevention of CHD can be achieved by improving medication adherence.

### Funding

This study was funded by the the Nursing Research Project of Chinese Medical Association Journal (ID: CMAPH-NRP2021008). The funders had no role in study design, data collection and analysis, decision to publish, or preparation of the manuscript.

### Grant Disclosures

The following grant information was disclosed by the authors:
Nursing Research Project of Chinese Medical Association Journal: CMAPH-NRP2021008.

### Competing Interests
The authors declare there are no competing interests.

### Author Contributions
- Jiaoyu Cao conceived and designed the experiments, authored or reviewed drafts of the article, and approved the final draft.
- Lixiang Zhang performed the experiments, analyzed the data, prepared figures and/or tables, and approved the final draft.
- Xiaojuan Zhou performed the experiments, prepared figures and/or tables, and approved the final draft.

### Data Availability
   The raw data are available in the Supplemental Files.

### Supplemental Information
Supplemental information for this article can be found online at http://dx.doi.org/10.7717/peerj.15876#supplemental-information.

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
