# Peer review of "Constructing a prognostic tool for predicting the risk of non-adherence to antiplatelet therapy in discharged patients with coronary heart disease: a retrospective cohort study"

_PeerJ, doi:10.7717/peerj.15876_

## Round 0.1 · original submission · Major Revisions

Please follow and address all comments of the reviewers.

·

Basic reporting

.

Experimental design

.

Validity of the findings

.

Additional comments

Non-adherence to anticoagulant therapy is of utmost importance among cardiology patients. Development of the tool for optimizing adherence, therefore, is crucial after discharge from the hospital. Unfortunately, the word “nomogram” might not be used properly in this context. If we use the dictionary definition, where nomogram is a graphic representation that consists of several lines marked off to scale and arranged in such a way that by using a straightedge to connect known values on two lines an unknown value can be read at the point of intersection with another line; then this is a valid word. Personally, I’d prefer to use “tool” or so….
1. Moreover, I do not feel that this title reflects the contents of the article, at all. It appears that authors surveilled patients’ non-adherence behavior of antiplatelet drug treatment, and as a result, they came up with a “nomogram”.
2. Using the net-benefit curve in this instance has the role exclusively in further developments of machine learning appliances, so this potential value should be highlighted adequately.
3. The conclusion section in the abstract is simply repeating the results, I miss the part where you actually conclude something.
4. “Limitations” have to be elaborated more thoroughly. You should not treat the same patients living alone, and those discharged to the family setting. Perhaps, consider using the paper of Marcatto et al., and Hylek – to understand better the importance of age in adherence to the therapy.
5. What does the line labeled “Statistic” in Table 1 mean?
6. When writing decimal numbers, instead of “ 0.890” it is enough to use “0.89” instead
7. of “stable angina pectoris” You might consider the abbreviation “SAP”; the “unstable angina pectoris” respectively; “acute myocardial infarction”… if necessary use footnotes – just try to reduce the size of a table.

·

Basic reporting

Relevant topic in the given geographical context and may also have wider applicability.

The first language of the authors is likely not English. Review by a fluent English-speaking person is recommended. Some of the sentences are worded in a confusing way, thereby limiting the usefulness to a wider audience (e.g., lines 80 to 84; lines 100 to 102; lines 259 to 263; lines 266 to 267; etc.).

Several sentences are extremely long (e.g., lines 181 to 186: 86 words; or lines 245 to 251: 83 words; or lines 307 to 312: 80 words; etc.). Very long sentences make it hard to follow what the intended message is. Please break up very long sentences into 2 or 3 or more shorter sentences in order to improve comprehension. Sentences of 15 to 20 words are the most readable but can be somewhat longer in scientific communications (e.g., up to thirty words or so).

Grammar and punctuation need to be scrutinized and corrected as required. There are multiple punctuation inaccuracies.

21 of the 24 citations are from Chinese (by name) authors. There might be merit in expanding the list to include a more international sample of references (if available), which would also increase the appeal of the contents to an international readership.

Experimental design

Extensive workup to make the topic applicable in a wider setting. It is a retrospective study; hence the conclusions are applicable to the intended audience. Some aspects of the details may not pertain to an international audience (e.g., patient payment methods) but are still valid in the original context.

Good tables with ample data.

Expressive figures; very useful for visual comprehension.

Validity of the findings

The findings are borne out by the statistical analysis. The conclusions are valid and have practical applicability. Similar studies could also be carried out in other parts of the world. Suggest follow-up review and expansion of the findings in a future article. Very good initiative. Congratulation to the authors.

Additional comments

The contents are good and provide valid data. The English language text in its current form detracts from the message the paper is meant to convey. Remember, content and form both matter.

---

## Round 0.2 · accepted · Accept

The authors properly addressed the comments and requests of the reviewers. Thank you for your scientific contribution.

·

Basic reporting

I agree with thisver sion

Experimental design

See 1)

Validity of the findings

I agree with this version

Additional comments

See 3)

·

Basic reporting

Clear, unambiguous, professional English. Extensive literature references. Appropriate medical background and context. Professional organization of the article including tables. Relevant results for this topic.

Experimental design

Primary research in compliance with the aims and scope of the journal. Research well-designed and clinically relevant, with wide patient applicability. Technical and ethical standards above reproach. Detailed and informative methodology.

Validity of the findings

Valid medical initiative, with the paper encouraging further studies in the field. The data as provided are sound and statistically verified. Conclusions relevant to the topic discussed, with supportive results.

Additional comments

Valid clinical initiative, relevant and applicable to a wide patient population. The paper encourages further research in this field, including internationally. Thank you for drawing attention to this very important worldwide clinical and public health matter.